# Review of Promising Off-Label Use of Deucravacitinib

**DOI:** 10.3390/ijms26199447

**Published:** 2025-09-27

**Authors:** Yoshihito Mima, Masako Yamamoto, Ken Iozumi

**Affiliations:** Department of Dermatology, Tokyo Metropolitan Police Hospital, Tokyo 164-8541, Japan

**Keywords:** tyrosine kinase 2, interleukin-17/23, type 1 interferons

## Abstract

Tyrosine kinase 2 (TYK2) mediates the signaling pathways of proinflammatory cytokines such as interleukin (IL)-12, IL-23, and type I interferons (IFNs) and plays a pivotal role in the pathogenesis of psoriasis and various other immune-mediated diseases. Deucravacitinib, a selective oral TYK2 inhibitor, has been approved for the treatment of psoriasis and demonstrated high efficacy and a favorable safety profile. This review summarizes the potential for expanding deucravacitinib indications based on case reports, clinical trials, and preclinical studies. Diseases in which TYK2 pathway has been demonstrated to be involved and for which clinical benefit of deucravacitinib has been reported include discoid lupus erythematosus, systemic lupus erythematosus, alopecia areata, lichen planus, palmoplantar pustulosis, psoriatic arthritis, systemic sclerosis, interstitial pneumonia, inflammatory bowel disease, and chronic recurrent multifocal osteomyelitis. Furthermore, emerging research suggests potential therapeutic applications in neurodegenerative diseases such as Alzheimer’s disease, and malignancies such as type 1 diabetes, vascular calcification in chronic kidney disease, T-cell acute lymphoblastic leukemia, and multiple sclerosis. Deucravacitinib may exert therapeutic effects by broadly suppressing cytokine signaling in a diverse range of inflammatory disorders. Ongoing clinical trials and mechanistic studies are required to clarify the efficacy and support its future indications.

## 1. Introduction

Tyrosine kinase (TYK) 2, a member of the Janus kinase (JAK) family, plays a critical role in the functions of immune and hematopoietic cells, including regulation of cell proliferation, survival, differentiation, and immune responses. TYK2 specifically mediates intracellular signaling of proinflammatory cytokines such as interleukin (IL)-12, IL-23, and type I interferon (IFN), which are implicated in the pathogenesis of psoriasis and other immune-mediated diseases [1]. Type I INFs signal through the JAK1–TYK2 complex, whereas IL-12 and IL-23 utilize the JAK2–TYK2 complex. TYK2 forms receptor complexes with other JAK molecules, and is essential for signal transduction [2]. Upon activation, JAKs phosphorylate cytokine receptors, creating docking sites for signal transducers and activators of transcription (STAT) proteins. STAT proteins then translocate to the nucleus, where they regulate immune activation and inflammatory responses [3]. In the IFN/Th1 pathway, naïve T cells differentiate into T helper (Th)1 cells, which produce IFNs to activate macrophages and enhance antigen presentation [4]. In contrast, the IL-23/Th17/IL-17 pathway promotes differentiation of naïve T cells into Th17 cells upon stimulation with IL-23. The Th17 cells secrete IL-17, which recruits neutrophils and drives inflammation [5]. Inflammatory pathways mediated by IFN, Th1, and Th17 cells are closely associated with the development of psoriasis and other inflammatory disorders [4,5,6].

### Psoriasis

Deucravacitinib, a TYK2 inhibitor, was approved in 2022 for moderate-to-severe plaque psoriasis who were candidates for systemic or phototherapy [7]. Psoriasis is a chronic, immune-mediated inflammatory skin disease that significantly impairs patients’ quality of life (QoL) [8,9]. Clinically, it typically presents as well-demarcated erythematous plaques with silvery white scales that commonly affect the scalp, buttocks, elbows, and knees [10]. Furthermore, psoriasis is associated with a wide range of systemic comorbidities, including psoriatic arthritis, cardiovascular disease, metabolic syndrome, obesity, diabetes, hypertension, autoimmune diseases, malignancies, inflammatory bowel disease, nonalcoholic fatty liver disease, and depression [11]. The pathogenesis of psoriasis is primarily driven by Th cells, particularly the Th1 and Th17 subsets [8]. IL-12 is essential for the differentiation and proliferation of Th1 cells and promotes the production of IFN and tumor necrosis factor-alpha (TNF-α). In contrast, IL-23 is critical for the maintenance and expansion of Th17 cells. Both IL-12 and IL-23 are considered key pathogenic mediators of plaque psoriasis [12]. Activated Th1 and Th17 cells secrete inflammatory cytokines such as IL-17A, IL-17F, IL-22, and TNF-α, which drive keratinocyte hyperproliferation and inflammation in both the epidermis and dermis [12,13,14].

In addition to injectable biologics targeting IL-17 and IL-23, which inhibit Th17-mediated keratinocyte activation, deucravacitinib has emerged as a promising oral therapeutic option. Deucravacitinib is a selective oral TYK2 inhibitor that blocks the signaling pathways of IL-12, IL-23, and type I IFNs by selectively inhibiting TYK2, which mediates the intracellular signaling of these cytokines. This inhibition disrupts cytokine-driven pathways, thereby suppressing keratinocyte hyperproliferation and inflammation, resulting in therapeutic efficacy against plaque psoriasis [15,16,17]. The mechanism of action of deucravacitinib fundamentally differs from that of conventional JAK inhibitors. Traditional JAK inhibitors target the highly conserved active kinase domain shared among the JAK family members, whereas deucravacitinib is a selective allosteric TYK2 inhibitor that specifically binds to the regulatory pseudo-kinase domain of TYK2. This binding induces a conformational change, locking TYK2 in an inactive state and thereby inhibiting downstream signaling [18]. This unique allosteric inhibition confers exceptional selectivity, with in vitro studies demonstrating 100- to 2000-fold selectivity for TYK2 over JAK1, JAK2, and JAK3 [19].

In the POETYK PSO-1 trial, which enrolled patients with moderate to severe plaque psoriasis, deucravacitinib demonstrated significantly greater efficacy than apremilast or placebo. At week 24, approximately 70% of patients receiving deucravacitinib achieved Psoriasis Area Severity Index (PASI) 75, and approximately 42% reached PASI 90 [16]. Similarly, in the Japan-based POETYK PSO-4 trial, which included a population with a generally lower baseline body mass index (BMI) than that included the POETYK PSO-1 trial, at week 24, over 80% of the patients achieved PASI 75 and approximately 60% achieved PASI 90 [20]. Beyond improvements in skin lesions, deucravacitinib also enhanced QoL, with approximately 60% of the patients achieving a Dermatology Life Quality Index (DLQI) score of 0 or 1 at week 24 [21]. Common adverse events of deucravacitinib include nasopharyngitis, upper respiratory tract infections, headache, diarrhea, and nausea. However, unlike traditional JAK inhibitors, deucravacitinib has a favorable safety profile. The incidence of serious infections, thromboembolic events, cytopenia, and clinically significant laboratory abnormalities was low. Most adverse events reported in clinical trials are mild to moderate in severity, and the rate of treatment discontinuation due to adverse events is low, indicating a high level of safety [16,20].

In addition to clinical trials, real-world evidence supports the efficacy of deucravacitinib in routine clinical practice. A retrospective analysis involving 33 patients with psoriasis demonstrated that approximately 80% and 60% achieved PASI 75 and 90, respectively, which is comparable to the outcomes of clinical trials [16,20,21]. Pruritus, which significantly impairs QoL in psoriasis, was also found to improve significantly within just one month of deucravacitinib initiation in this cohort [22,23]. Furthermore, retrospective analyses involving cohorts of approximately 70–100 patients have identified clinical predictors of treatment response [24,25]. Specifically, in two retrospective studies evaluating outcomes at weeks 16 and 52, patients with BMI < 25 demonstrated significantly better responses compared with those with BMI ≥ 25, and patients aged ≥ 65 years responded more favorably than younger patients [24,25]. Interestingly, however, the achievement rate of PASI 100 was slightly lower in patients aged ≥ 65 years [25]. Indeed, a retrospective analysis of predictors of complete skin clearance (PASI clear) at week 52 revealed that BMI and age were key determinants [24]. Receiver operating characteristic curve analysis identified BMI < 22.9 kg/m^2^ and age < 61 years as predictive factors for achieving PASI clear, and patients meeting these criteria experienced earlier and greater improvements in quality of life. Taken together, while younger patients exhibited less favorable treatment responsiveness compared with older patients, they were more likely to achieve PASI clear [24]. Notably, even among patients with inadequate responses to prior biological therapies, approximately two-thirds achieved PASI 75 after switching to deucravacitinib [26]. However, it is important to note that the PASI response rates for deucravacitinib tended to be lower than those for biological agents. Moreover, prior exposure to biologics has been associated with reduced treatment responsiveness to deucravacitinib [27,28]. Of particular interest, deucravacitinib has demonstrated efficacy in traditionally difficult-to-treat areas, such as the scalp, palms, soles, and nails, which often show a limited response to conventional therapies. Subgroup analyses from the POETYK PSO-4 trial revealed improvements at these challenging sites comparable to those observed in the trunk and extremities. These findings have been corroborated in real-world retrospective studies [29,30].

Although the therapeutic efficacy of deucravacitinib in psoriasis has been demonstrated in clinical trials and its real-world effectiveness has been gradually validated in routine clinical practice, its approved indication remains limited to psoriasis. TYK2 mediates the intracellular signaling of a broad range of cytokines involved in both innate and adaptive immune responses, including those implicated in psoriasis, such as IL-12, IL-23, IL-6, IL-10, IL-13, IL-27, and IL-31. These findings suggested that TYK2 plays a central role in the pathogenesis of various immune-mediated diseases [2,18,31]. Although deucravacitinib is currently approved for the treatment of psoriasis, its ability to broadly inhibit cytokine signaling makes it a promising therapeutic candidate for a wide array of immune-mediated conditions.

A review by Bang et al. discussed the potential future indications for deucravacitinib beyond psoriasis [32]. Emerging evidence from preclinical models and genetic studies has highlighted the association between TYK2 variants and several immune-related diseases. Research in these areas is advancing rapidly. In this review, we aim to comprehensively explore the therapeutic potential of deucravacitinib based on these findings.

## 2. Review

This study offers a comprehensive review of potential disease targets of deucravacitinib (Graphical abstract). Diseases suggesting the therapeutic potential of deucravacitinib in case reports are shown in Table 1. Ongoing clinical trials of deucravacitinib in off-label use are shown in Table 2.

### 2.1. Discoid Lupus Erythema

Discoid lupus erythematosus (DLE) is a scarring form of cutaneous lupus erythematosus characterized by inflammation, pigmentary changes, and alopecia. Due to its often disfiguring nature, DLE can have a profound impact on a patients’ QoL. In conventional treatment, first line treatment is primarily antimalarial agents and topical or intralesional corticosteroids, whereas the second- and third-line options include mycophenolate mofetil, methotrexate, and thalidomide or lenalidomide. Recently, evidence supporting the efficacy of anifrolumab has emerged [33,34]. Ezeh et al. reported a case of erythema-predominant DLE that was refractory to topical corticosteroids, calcineurin inhibitors, intralesional triamcinolone, and oral hydroxychloroquine. The patient was treated with 6 mg/day deucravacitinib in combination with hydroxychloroquine (400 mg/day). Remarkable improvement in erythema with scaling, skin pain, and pruritus was observed within one month of initiation [35]. Similarly, Aw et al. described a patient with treatment-resistant DLE, characterized by inflammatory scarring alopecia, who was administered deucravacitinib (6 mg/day). Six months after starting therapy, the patient exhibited significant hair regrowth and lesions [36]. TYK2 is involved in the signal transduction of multiple cytokines, including IL-10, IL-12, IL-23, and type I IFN, of which IL-12 and type I IFN play key roles in the pathogenesis of systemic lupus erythematosus (SLE) and DLE [37]. By selectively inhibiting TYK2, deucravacitinib may block the signaling of these proinflammatory cytokines, thereby attenuating the inflammation and immune dysregulation observed in DLE [38]. These case reports suggest the utility of deucravacitinib in managing refractory DLE, particularly in cases complicated by scarring alopecia, and highlight the need for further clinical investigations to validate these findings [35,36].

### 2.2. Systemic Lupus Erythematosus

Systemic lupus erythematosus (SLE) is a chronic autoimmune disease characterized by the production of antinuclear antibodies and has a wide spectrum of clinical manifestations [39]. Its pathogenesis involves a complex interplay between genetic predisposition, environmental factors, and dysregulated immune responses, production of autoantibodies (e.g., antinuclear antibodies), and immune complex formation, which play a central role in tissue damage [40]. The clinical manifestations of SLE range from relatively mild symptoms such as malar rash and arthralgia to severe organ involvement, including lupus nephritis, which is a major cause of morbidity and mortality [41]. Key cytokines implicated in SLE pathogenesis include type I IFN, IL-10, IL-12, and IL-23. These cytokines signal via the intracellular TYK2 [19,42,43]. Deucravacitinib, a selective TYK2 inhibitor, has been shown to strongly suppress signaling pathways mediated by these cytokines in human cells [19]. Its efficacy may be comparable to or even exceed that of the currently approved biologics targeting type I IFN receptors for SLE. In a phase 2 clinical trial of deucravacitinib in patients with SLE, the primary endpoint—SLE responder index (SRI)-4 response rate—was significantly higher in the deucravacitinib group (49.5%) than in the placebo group (34.4%), with an odds ratio of 1.9 (95% confidence interval: 1.0–3.4; *p* = 0.02) [44,45,46]. Furthermore, deucravacitinib showed statistically significant improvements over the placebo across multiple secondary endpoints, such as outcomes of joint assessments. Biomarkers associated with SLE, such as anti-dsDNA antibody titers and complement components C3 and C4, also significantly improved in the deucravacitinib group [46]. A systematic review and network meta-analysis comparing the efficacy of treatments for SLE revealed that deucravacitinib achieved a significantly higher odds of response in CLASI-50 (a 50% reduction in the Cutaneous Lupus Erythematosus Disease Area and Severity Index) compared with placebo (odds ratio: 8.28; 95% CI: 2.22–30.91). Notably, deucravacitinib also outperformed existing type I IFN-targeted therapies, such as litifilimab and anifrolumab [47].

SLE remains a challenging relapsing disease with a high risk of organ damage and premature mortality. However, these clinical trial results and comparative analyses suggest that deucravacitinib is a promising therapeutic option. Further validation may help reduce organ involvement and improve long-term outcomes in patients with SLE [46,47,48,49].

### 2.3. Alopecia Areata

Alopecia areata (AA) is a non-scarring hair loss disorder caused by autoimmune-mediated inflammation that affects hair follicles. Under normal conditions, hair follicles are the immune-privileged sites, i.e., they are protected against autoimmune attacks. However, in AA, immune tolerance breaks down, leading to the activation of cluster of differentiation (CD)8-positive T cells (particularly NKG2D subsets) and CD4-positive T cells, which recognize hair follicles as self-antigens and attack. These T cells release Th1-associated cytokines such as IFNs via the JAK-STAT signaling pathway, further amplifying the immune response. In addition to immune dysregulation, genetic susceptibility, such as polymorphisms in human leukocyte antigen (HLA) genes and cytotoxic T-lymphocyte associated protein 4 (CTLA4), and environmental factors, including stress and infections, have been implicated as potential triggers. This pathological immune activity causes a premature transition of hair follicles from the growth phase to the regression phase, leading to disruption of the hair cycle and subsequent hair loss [50,51,52,53,54].

Oliel et al. reported a case of AA in a patient with concomitant psoriasis that was refractory to topical steroids and minoxidil. The patient was treated with deucravacitinib, a TYK2 inhibitor, at a dose of 6 mg/day. After six weeks of treatment, the Severity of Alopecia Tool (SALT) score improved markedly from 97 to 45.2 [55].

IFNs and IL-15 have been identified as key pathogenic drivers of AA, making the JAK/STAT pathway an attractive therapeutic target [56]. Baricitinib (a JAK1/2 inhibitor) and ritlecitinib (a JAK3/TEC family kinase inhibitor) have been approved for patients with AA who are unresponsive to topical treatments [57]. Furthermore, improvements in SALT scores following ritlecitinib therapy have been shown to positively correlate with reductions in Th1-associated markers, highlighting the central role of Th1 inflammation in disease activity [58]. Deucravacitinib selectively inhibits TYK2 by binding to its regulatory pseudokinase domain, thereby suppressing signal transduction mediated by cytokines such as IFN. This mechanism may attenuate Th1-driven inflammation and contribute to the clinical improvement of AA [37,55,56].

### 2.4. Lichen Planus

Lichen planus (LP) is an inflammatory disorder characterized by pruritic, violaceous, flat-topped, shiny papules that affect the skin, oral mucosa, genital mucosa, and other sites. Its etiology is multifactorial and involves immune dysregulation, hepatitis C virus infection, medications, and psychological stress [59]. Stolte et al. reported three patients with oral LP resistant to topical corticosteroids who were administered 6 mg/day deucravacitinib. After 12 weeks of treatment, all three patients exhibited significant improvements in both erosive and reticular lesions [60]. The pathogenesis of LP is thought to involve CD4 and CD8-positive T cells infiltrating the dermis, as well as proinflammatory cytokines and cytotoxic mediators secreted by plasmacytoid dendritic cells, leading to the apoptosis of basal keratinocytes [61]. IFNs, IL-12, and IL-23 are key cytokines implicated in LP that contribute to basal layer inflammation and disease progression [62,63]. Deucravacitinib inhibits TYK2 activity by binding to its regulatory pseudokinase allosteric domain, thereby suppressing IL-12, IL-23, IFN, and IL-17 production. This mechanism may attenuate the inflammatory processes that drive LP pathogenesis [37,60]. Immunohistochemical analyses of the aforementioned cases revealed reduced IL-12, IL-23, IFN, and IL-17 expression following deucravacitinib treatment, which is consistent with the proposed mode of action [60]. These findings suggest that targeting the TYK2 signaling pathway with deucravacitinib is a promising therapeutic option for patients with LP who are unresponsive to conventional topical therapies [60,61,62,63,64].

### 2.5. Palmoplantar Pustulosis

Palmoplantar pustulosis (PPP) is a chronic and relapsing inflammatory skin disorder characterized by erythematous and scaly lesions on the palms and soles, followed by the development of numerous sterile pustules [65]. Its pathogenesis involves a complex interplay between immune dysregulation, environmental factors, and genetic predisposition. Inflammation is predominantly centered around the intraepidermal eccrine sweat ducts, with contributions from Langerhans cells and proinflammatory cytokines such as IL-8, IL-1α, IL-1β, IL-17, IL-22, and IL-23. Elevated serum levels of TNF-α, IL-17, IL-22, and IFN suggest the involvement of systemic inflammation. Additionally, nicotine is thought to contribute to disease development by acting on the eccrine glands or keratinocytes and promoting hyperkeratosis and neutrophil-driven inflammation [66,67]. De Luca et al. reported a case series involving five patients with PPP treated with deucravacitinib, a selective TYK2 inhibitor. Of these, three patients showed clinical improvement at 16 weeks. Deucravacitinib is believed to exert its therapeutic effects by suppressing the production of key pathogenic cytokines involved in PPP, including IL-23, IFN, and IL-17. However, all five patients experienced transient worsening of skin lesions at week 4, highlighting the need for further investigation and accumulation of clinical evidence to establish its efficacy in PPP [68].

### 2.6. Psoriatic Arthritis

Psoriatic arthritis (PsA) is a chronic inflammatory joint disease that affects approximately 30% of patients with psoriasis [69]. PsA is a heterogeneous disorder that primarily involves the peripheral joints but may also affect the axial skeleton and entheses. The most common clinical subtype of PsA is peripheral polyarthritis, which resembles rheumatoid arthritis and asymmetric oligoarthritis [70]. Other recognized phenotypes include monoarthritis, distal interphalangeal-predominant arthritis, axial disease, and arthritis mutilans, a severe form of arthritis characterized by bone resorption and joint destruction. Many patients also present with dactylitis and nail changes, such as pitting, subungual hyperkeratosis, onycholysis, and oil-drop discoloration [71]. The pathogenesis of PsA involves a complex interplay between genetic and environmental factors, triggering immune activation and leading to the production of proinflammatory cytokines, including TNF-α, IL-17, and IL-23. The IL-23–IL-17 signaling axis is considered a central pathogenic pathway shared with psoriasis [13,72,73]. Currently approved therapies for PsA include TNF-, IL-17, IL-23, and JAK inhibitors. Deucravacitinib, a selective TYK2 inhibitor, has been suggested to be effective against PsA by suppressing IL-17 and IL-23 production [74]. In a phase 2 clinical trial, deucravacitinib demonstrated a significantly higher rate of minimal disease activity (MDA) at week 16 than the placebo (23.9% vs. 7.6%) [75]. MDA was defined as achieving at least five of the following seven criteria: tender joint count (TJC) ≤ 1, swollen joint count (SJC) ≤ 1, Patient Global Assessment of Pain (Pain VAS) ≤ 15, Patient Global Assessment of Disease Activity (PtGA) ≤ 20, Health Assessment Questionnaire–Disability Index (HAQ-DI) ≤ 0.5, PASI score ≤ 1 or body surface area (BSA) ≤ 3%, and ≤1 tender entheseal point as measured using the Leeds Enthesitis Index (LEI). Furthermore, in the same phase 2 study, the proportion of patients achieving an ACR20 response at week 16 was significantly higher in the deucravacitinib group (52.9%) than that in the placebo group (31.8%) [76]. These findings suggest that targeting the TYK2 pathway with deucravacitinib is a promising therapeutic option for PsA, complementing existing treatments beyond conventional topical or systemic therapies [13,69,70,71,72,73,74,75,76].

### 2.7. Systemic Sclerosis

Systemic sclerosis (SSc) is a chronic fibrotic disease driven by autoimmune mechanisms that affect not only the skin but also multiple internal organs. Clinically, it is characterized by marked skin sclerosis involving the extremities and face, often preceded by Raynaud phenomenon. Digital ulcers may develop as disease progresses, leading to necrosis or distal digit loss. Other common features include extensive subcutaneous calcinosis, severe pruritus, and telangiectasia [77,78,79]. The pathogenesis of SSc is classically divided into three interconnected stages: (1) endothelial dysfunction, (2) immune dysregulation, and (3) tissue fibrosis, all of which are associated with distinct histopathological features. Initially, endothelial cell swelling occurs, followed by the perivascular infiltration of lymphocytes and histiocytes. Subsequently, activated myofibroblasts proliferate, accompanied by excessive deposition of extracellular matrix (ECM) components, particularly homogenized collagen bundles [80,81]. Ongoing research on the cellular and molecular mechanisms of SSc has shed light on potential therapeutic targets. SSc arises from a complex imbalance between the innate and adaptive immune systems that is underpinned by genetic susceptibility. This leads to the production of a wide array of cytokines, chemokines, and autoantibodies that promote fibroblast activation and differentiation into myofibroblasts, resulting in excessive fibrotic tissue formation. Although curative therapy is currently lacking, significant advances have been made in symptomatic and organ-specific management [81]. Key profibrotic cytokines implicated in SSc include IL-4, IL-13, and transforming growth factor-beta (TGF-β), which are secreted by Th2 cells, macrophages, and mast cells. These cytokines promote fibroblast activation and the overproduction of collagen and ECM. Th17-mediated inflammation contributes to dermal fibrosis and disease progression. M2 macrophages and neutrophils further exacerbate fibrosis through the release of reactive oxygen species and neutrophil extracellular traps. IFNs produced by dendritic cells and cytokines such as IL-6 and TGF-β secreted by aberrantly activated B cells also play important roles in fibrogenesis [82,83,84,85,86,87,88]. Early endothelial injury is considered a central trigger for immune dysregulation in SSc, and early infiltration of T cells and macrophages is commonly observed. Notably, cytokine expression is closely correlated with disease activity [77,78,79,80,81,82,83,84,85,86]. Deucravacitinib, a selective TYK2 inhibitor, blocks downstream signaling of multiple cytokines implicated in SSc, including IL-6, IL-13, and type I IFNs. This mechanism may attenuate inflammation and fibrosis [89]. Fukasawa et al. reported three cases of SSc with coexisting psoriasis treated with deucravacitinib for six months. In addition to the improvement in psoriatic lesions, all patients exhibited a reduction in six or more points in the modified Rodnan total skin thickness score. Furthermore, a decrease in the proportion of peripheral blood Th17 and B cells was observed, suggesting that deucravacitinib suppressed the activation of these immune cell subsets, thereby mitigating both inflammation and fibrosis in SSc. Although these observations are limited to case reports, they highlight the potential of deucravacitinib as a novel therapeutic agent for SSc. Further large-scale clinical trials are warranted to evaluate the efficacy and safety of this context [89]. However, its efficacy has only been suggested in case series, and the evidence remains preliminary, which constitutes a key limitation.

### 2.8. Inflammatory Bowel Diseases (Crohn’s Disease and Ulcerative Colitis)

Inflammatory bowel diseases (IBD), such as Crohn’s disease (CD) and ulcerative colitis (UC), are chronic and relapsing systemic immune-mediated inflammatory disorders that lead to intestinal damage [90]. The pathogenesis of IBD involves a complex interplay between genetic susceptibility and environmental factors, with inflammation primarily mediated by the effector cells of both the innate and adaptive immune systems [91]. In CD, Th1 and Th17 responses are predominantly accompanied by impaired regulatory T-cell function, whereas UC is mainly associated with Th2 and Th17 responses [92,93,94,95]. TYK2 forms a heterodimer with JAK2 and is involved in the downstream signaling of the IL-12 and IL-23 receptors. IL-12 promotes Th1 cell differentiation and the production of proinflammatory cytokines such as TNF-α and IFNs, while IL-23 supports Th17 cell proliferation and survival. Therefore, TYK2 inhibition may suppress both Th1- and Th17-mediated inflammation, thus offering a potential strategy for controlling chronic inflammation in CD and UC [92,93,94,95].

IL-23 inhibitors, such as guselkumab and risankizumab, which target the IL-23/IL-17 axis, have shown significant clinical benefits in moderate-to-severe CD and UC and are now approved for the treatment of refractory IBD [96,97,98]. In a phase 3, multicenter, open-label, randomized trial, patients with moderate-to-severe Crohn’s disease who had either an inadequate response to or were unable to continue anti-TNF therapy due to adverse events were assigned to receive risankizumab or ustekinumab for 48 weeks. The co-primary endpoints were clinical remission at week 24 (defined as CDAI < 150) and endoscopic remission at week 48 (defined as SES-CD ≤ 4, a ≥2-point reduction from baseline, and no individual subscore > 1). Risankizumab was non-inferior to ustekinumab for clinical remission at week 24 (58.6% vs. 39.5%) and demonstrated superiority for endoscopic remission at week 48 (31.8% vs. 16.2%, *p* < 0.001). The incidence of adverse events was similar between the two groups. These findings confirm that risankizumab is non-inferior to ustekinumab in achieving clinical remission at week 24 and superior in achieving endoscopic remission at week 48 in patients with Crohn’s disease who are refractory or intolerant to TNF-α inhibitors [97]. In a phase 3 randomized, double-blind, placebo-controlled trial, the efficacy and safety of guselkumab as induction and maintenance therapy in patients with ulcerative colitis were evaluated. A total of 701 patients were enrolled in the induction trial, with 421 (60%) randomized to receive intravenous guselkumab 200 mg and 280 (40%) to placebo. At week 12, the clinical remission rate was significantly higher in the guselkumab group than in the placebo group (23% [95/421] vs. 8% [22/280]; difference, 15%; 95% CI, 10–20; *p* < 0.0001). The maintenance trial included 568 induction responders, who were randomized to subcutaneous guselkumab 200 mg every 4 weeks (190 [33%]), guselkumab 100 mg every 8 weeks (188 [33%]), or placebo (190 [33%]). At week 44, clinical remission was achieved by 50% of patients in the 200 mg group (95/190; difference vs. placebo, 30%; 95% CI, 21–38; *p* < 0.0001) and 45% in the 100 mg group (85/188; difference vs. placebo, 25%; 95% CI, 16–34; *p* < 0.0001), compared with 19% (36/190) in the placebo group. The overall safety profile was favorable and consistent with prior experience in approved indications. In the induction trial, adverse events occurred in approximately 49% of patients in both groups (guselkumab, 208/421; placebo, 138/280), with serious adverse events reported in 3% (12/421) of guselkumab-treated patients and 7% (20/280) of placebo-treated patients. Adverse events leading to treatment discontinuation occurred in 2% (7/421) and 4% (11/280), respectively. These results demonstrate that guselkumab is both effective and safe as induction and maintenance therapy for patients with moderate-to-severe active ulcerative colitis [98]. Additionally, JAK inhibitors such as upadacitinib and tofacitinib have been approved for UC. Deucravacitinib suppresses not only the IL-23/IL-17 pathway, but also Th1-related cytokines, making it a promising therapeutic candidate for both CD and UC [92,93,94,95,99,100]. Genome-wide association studies have identified the TYK2 locus as a susceptibility locus for IBD, further supporting its role as a therapeutic target for intestinal inflammation [101,102]. To date, three randomized, double-blind, placebo-controlled phase 2 trials have evaluated deucravacitinib in IBD: LATTICE-CD (NCT03599622) for patients with moderate-to-severe CD, and LATTICE-UC (NCT03934216) and IM011-127 (NCT04613518) for patients with UC. In these trials, patients were randomized to receive 3, 6, or 12 mg deucravacitinib twice daily, or a placebo for 12 weeks. The primary endpoints were clinical remission and endoscopic response at week 12 in LATTICE-CD, clinical remission based on the modified Mayo score at week 12 in LATTICE-UC, and clinical response at week 2 in IM011-127. In all three studies, the primary endpoints were not statistically significant. However, in the 12 mg twice-daily group, while the clinical response rate was similar between the deucravacitinib and placebo groups (53.8% vs. 50.0%), endoscopic remission, a more stringent outcome, was achieved in 28.0% of the deucravacitinib group versus 0% in the placebo group, suggesting efficacy. However, these findings must be interpreted with caution due to the limited sample size and insufficient statistical power [103]. While the mechanism of action supports the use of deucravacitinib in IBD, its clinical efficacy has not yet been demonstrated to a statistically significant degree compared to placebo [92,93,94,95,99,100,101,102,103]. Notably, IL-23 inhibitors are typically administered at higher doses to patients with IBD than to those with psoriasis, raising the possibility that the doses used in these trials may have been suboptimal. Therefore, future studies with optimized dosing regimens are warranted [103]. Importantly, the safety profile of deucravacitinib in these trials was favorable and consistent with the findings of previous psoriasis studies. For elderly patients or those at risk for adverse events associated with broader JAK inhibition, deucravacitinib may be a safer and more targeted therapeutic option. The potential for broader clinical applications in IBD remains an area of active interest for future investigation.

### 2.9. Interstitial Pneumonia

Interstitial pneumonia (IP) is a heterogeneous group of disorders characterized by inflammation and fibrosis of the lung parenchyma. These conditions often lead to progressive respiratory symptoms, impaired pulmonary function, respiratory failure, and a marked reduction in QoL [104]. The etiologies of IP include connective tissue diseases, environmental exposure, drug-induced lung injury, radiation therapy, occupational hazards, and allergens [105]. When lung injury occurs, growth factors such as TGF-β are released, promoting fibroblast activation and excessive proliferation. This process results in the abnormal accumulation of ECM, including collagen, within the pulmonary interstitium, driving tissue remodeling and irreversible structural changes [106]. The pathogenesis of IP involves a complex interplay between immune cells, cytokines, and growth factors. In particular, proinflammatory and profibrotic cytokines such as IL-6, IL-13, IL-23, and IL-31, along with TGF-β, are known to play pivotal roles [2,106,107,108,109,110]. Th2 cells contribute to fibrosis by secreting IL-13 and IL-31, which activate myofibroblasts and enhance collagen production. Meanwhile, Th17 cells also cooperate with TGF-β to produce IL-17, thereby indirectly promoting fibrotic progression [15]. IL-6 not only exacerbates local inflammation but also enhances TGF-β activity and promotes Th17 differentiation, further driving tissue remodeling and fibrosis [106,107,108,109]. TYK2 is a key intracellular signaling molecule that mediates the downstream signaling of proinflammatory and profibrotic cytokines, including IL-6, IL-13, IL-23, and IL-31 [110]. Deucravacitinib, a selective TYK2 inhibitor, has the potential to suppress chronic T-cell-mediated inflammation and fibrosis by blocking signaling pathways that include these cytokines [2,15,106,107,108,109,110]. Mima et al. recently reported a case of IP associated with psoriasis in which deucravacitinib administration improved interstitial lung abnormalities on chest computed tomography. In addition, immunological biomarkers reflecting IP severity and progression showed improvement [111]. Similarly, other JAK inhibitors that broadly suppress multiple cytokine pathways have been explored as therapeutic options for IP, with some reports demonstrating their clinical benefits [112]. Taken together, these findings suggest that deucravacitinib, by selectively and broadly inhibiting key cytokine signaling pathways, is a novel and promising therapeutic option for the IP management [2,15,104,105,106,107,108,109,110,111,112]. However, its efficacy has only been suggested in a case report, and the evidence remains preliminary, representing a key limitation.

### 2.10. Chronic Recurrent Multifocal Osteomyelitis

Chronic recurrent multifocal osteomyelitis (CRMO) is a rare, chronic autoinflammatory bone disorder that, if left untreated, can lead to progressive bone destruction. It primarily affects children and adolescents and is often associated with other immune-mediated conditions, such as IBD, severe acne vulgaris, ankylosing spondylitis, and psoriasis, including PPP. CRMO typically involves the metaphyses of long bones, particularly the femur and tibia. However, lesions may also occur in the pelvis, spine, clavicle, and mandible [113,114]. Although its precise pathophysiology remains unclear, CRMO is non-infectious, and accumulating evidence suggests that an imbalance in cytokine signaling plays a central role. Specifically, reduced expression of the anti-inflammatory cytokine IL-10 and overproduction of proinflammatory cytokines such as IL-1β, IL-6, and TNF-α have been reported in CRMO [115,116,117]. Nonsteroidal anti-inflammatory drugs (NSAIDs) are considered the first-line therapy for CRMO. In refractory cases, other treatment options include methotrexate, sulfasalazine, bisphosphonates, short-term glucocorticoids, and TNF-α inhibitors [118]. Biological agents that target IL-1, IL-6, IL-17, and IL-23 have recently emerged as promising therapeutic alternatives [119]. Glatzel et al. reported a case of CRMO that was resistant to both TNF-α and IL-17 inhibitors, in which the patient achieved complete clinical and radiological remission following treatment with deucravacitinib [120]. TYK2 mediates the signaling of various cytokines involved in osteoclastogenesis, including IL-6, IL-11, and IL-23 [121]. These findings suggest that the selective inhibition of TYK2 by deucravacitinib is a novel and effective therapeutic strategy for CRMO.

**Table 1 ijms-26-09447-t001:** Diseases for which case reports have suggested the therapeutic potential of deucravacitinib.

Disease	Evidence	The Suggested Mechanism	Article
Discoid lupus erythematosus	Case report (2 patients)	The inhibition of IL-12 and type 1 IFN	Ezeh N et al. [35] Aw K et al. [36]
Alopecia areata	Case report (1 patient)	The inhibition of type 1 IFN	Oliel S et al. [55]
Lichen planus	Case report (3 patients)	The inhibition of IL-12/17/23 and type 1 IFN	Stolte KN et al. [60]
Palmoplantar pustulosis	Case report (5 patients)	The inhibition of Il-17/23 and type 1 IFN	De Luca DA et al. [68]
Systemic sclerosis	Case report (3 patients)	The inhibition of IL-6/13/17/23	Fukasawa T et al. [89]
Interstitial pneumonia	Case report (1 patient)	The inhibition of IL-6/13/17/23	Mima Y et al. [111]
Chronic recurrent multifocal osteomyelitis	Case report (1 patient)	The inhibition of IL-1/6/17/23	Glatzel C et al. [120]

Abbreviations; IL: Interleukin; IFN: Interferon.

**Table 2 ijms-26-09447-t002:** Ongoing clinical trial indications for deucravacitinib.

Disease	Trial Registration	Clinical Trial	Patients	Response
SLE	NCT03252587	Phase 2	Active SLE (n = 363)	At week 32, SRI-4 response rates were 58% with deucravacitinib 3 mg BID (OR 2.8 [95% CI 1.5–5.1]; *p* < 0.001), 50% with 6 mg BID (OR 1.9 [1.0–3.4]; *p* = 0.02), and 45% with 12 mg QD (OR 1.6 [0.8–2.9]; nominal *p* = 0.08), versus 34% with placebo. By week 48, secondary endpoints—including BICLA response, CLASI-50, achievement of LLDAS, and joint scores—were consistently improved with deucravacitinib compared with placebo [46].
	NCT03920267	Phase 2	Active SLE (n = 261)	Ongoing
	NCT05617677	Phase 3	Active SLE (POETYK SLE-1) (n = 490)	Ongoing
	NCT05620407	Phase 3	Active SLE (POETYK SLE-2) (n = 490)	Ongoing
	NCT06875960	Open-label	SLE or DLE completing NCT03920267 or NCT03252587 (n = 35)	Ongoing
AA	NCT05556265	Phase 2	Active AA (n = 94)	Ongoing
PsA	NCT03881059	Phase 2	Active PsA (n = 203)	At week 16, ACR20 response rate was 52.9% with 6 mg (*p* = 0.0134) and 62.7% with 12 mg (*p* = 0.0004) versus 31.8% with placebo. Both doses also significantly improved HAQ-DI, SF-36 PCS, and PASI 75 compared with placebo (all *p* ≤ 0.05) [76].
	NCT04908189	Phase 3	Active PsA naïve to biologic disease modifying anti-rheumatic drugs or TNFα inhibitor treatment (n = 729)	Ongoing
	NCT04908202	Phase 3	Active PsA naïve to biologic disease-modifying anti-rheumatic drugs (n = 670)	Ongoing
	NCT06869551	Phase 3	Children and adolescents with active PsA (n = 60)	Ongoing
IBD	NCT03599622	Phase 2	Moderately to severe CD (n = 239)	Clinical remission at week 12 was 31.4% with deucravacitinib 3 mg, 19.0% with 6 mg, and 28.3% with placebo (*p* = 0.68 and *p* = 0.21 vs. placebo, respectively), indicating no significant between-group difference. Endoscopic response was higher with 3 mg than with placebo (23.3% vs. 8.3%; *p* = 0.02), whereas the 6 mg group did not differ from placebo (16.7% vs. 8.3%; *p* = 0.16). Clinical response rates were 48.8% and 39.3% with 3 mg and 6 mg, respectively, versus 38.3% with placebo; PRO2 remission occurred in 32.6% (*p* = 0.28) and 20.2% (*p* = 0.54) for the 3 mg and 6 mg groups, respectively, compared with 25.0% with placebo [103].
	NCT03934216	Phase 2	Moderate to severe UC (n = 131)	The week-12 clinical response rate was 53.8% with deucravacitinib versus 50.0% with placebo, and the primary endpoint was not met. Clinical remission occurred in 20.8% and 25.0%, respectively. Endoscopic improvement was observed in 32.0% versus 37.5%, whereas endoscopic remission was achieved in 28.0% of deucravacitinib-treated patients and 0% of those receiving placebo [103].
	NCT04613518	Phase 2	Moderate to severe UC (n = 38)	Clinical remission was achieved in 14.8% of patients receiving deucravacitinib versus 16.3% receiving placebo (*p* = 0.59), with no significant between-group difference. Clinical response occurred in 37.5% vs. 32.6% (*p* = 0.31) and endoscopic response in 19.3% vs. 27.9% (*p* = 0.88), neither reaching statistical significance [103].
	NCT04877990	Open-label	Previous participants in a Deucravacitinib Phase 2 study for CD or UC (n = 67)	Ongoing

Abbreviations; SLE: systemic lupus erythematosus, BID: twice daily, OR: odds ratio, QD: once daily, BICLA: BILAG-based Composite Lupus Assessment, CLASI: Cutaneous Lupus Erythematosus Disease Area and Severity Index, LLDAS: lupus low disease activity state, DLE: discoid lupus erythematosus, AA: alopecia areata, PsA: psoriatic arthritis, ACR: American College of Rheumatology, HAQ-DI: Health Assessment Questionnaire–Disability Index, SF-36: Short Form 36, PCS: Physical Component Summary (of SF-36), PASI: Psoriasis Area and Severity Index, TNF: tumor necrosis factor, IBD: inflammatory bowel disease, UC: ulcerative colitis, CD: Crohn’s disease, PRO2: 2-component patient-reported outcome based on stool frequency and rectal bleeding subscores in ulcerative colitis.

### 2.11. Others

Next, in Table 3, we summarize diseases beyond those in which the efficacy of deucravacitinib has been suggested by clinical trials or case series, focusing on conditions where experimental studies using human cells or mouse models have demonstrated upregulated TYK2 expression and provided evidence supporting the therapeutic potential of TYK2 inhibition.

Neuroinflammation is increasingly recognized as a fundamental mechanism underlying the pathogenesis of numerous neurodegenerative disorders, including Alzheimer’s disease and amyotrophic lateral sclerosis (ALS), and has emerged as a promising therapeutic target [122,123,124]. Cytoplasmic double-stranded RNA (cdsRNA) triggers a type I IFN-mediated innate immune response in human neurons, ultimately leading to cell death [125]. In the brains of patients with Alzheimer’s disease, cdsRNA colocalizes with pTDP-43 inclusions, and the affected regions show robust upregulation of IFN-responsive genes. JAK inhibitors, such as baricitinib and ruxolitinib, have demonstrated protective effects in specific cortical regions. CRISPR-based genetic screening identified TYK2, a member of the JAK family, as a key mediator of cdsRNA-induced neuronal toxicity. Treatment with deucravacitinib, a selective TYK2 inhibitor, effectively mitigated this cytotoxic effect. CCL2, CXCL10, and IL-6 have also been proposed as potential biomarkers of cdsRNA-related neurodegeneration, suggesting a shared neuroinflammatory mechanism between the TDP-43-associated forms of Alzheimer’s disease and ALS, with TYK2 emerging as a novel therapeutic target [126]. Further genetic screening identified TYK2 as a potential regulator of tau protein expression. TYK2 phosphorylates tau at tyrosine 29 (Tyr29), thereby stabilizing and promoting its aggregation in human cells. TYK2-dependent Tyr29 phosphorylation was also found to inhibit autophagic degradation of tau. In P301S tau-transgenic mouse models, TYK2-mediated phosphorylation led to pathogenic tau accumulation, whereas TYK2 knockdown reduced total and pathogenic tau levels and ameliorated gliosis. These findings suggest that partial inhibition of TYK2 is a potential promising therapeutic strategy in Alzheimer’s disease or ALS, a disease characterized by pathological tau hyperphosphorylation [127,128,129].

In chronic kidney disease (CKD), medial arterial calcification is linked to inflammation driven by hyperphosphatemia [130]. Multiple members of the IL-6 cytokine family contribute to pro-calcific processes in vascular smooth muscle cells (VSMCs), among which leukemia inhibitory factor (LIF) plays a central role [131,132]. LIF expression increases in response to phosphate exposure, further promoting calcification. Conversely, silencing of endogenous LIF or its receptor (LIFR) attenuates phosphate-induced calcification, whereas soluble LIFR exerts antagonistic effects. Mechanistically, LIF induces TYK2 and STAT3 in VSMCs [133]. Deucravacitinib, a selective allosteric TYK2 inhibitor approved for the treatment of psoriasis, suppresses LIF signaling and phosphate-induced calcification. In contrast, TYK2 overexpression exacerbates calcification. In mouse aortic ring models, both pharmacological inhibition and genetic deletion of Tyk2 significantly reduced vascular calcification, and similar findings were observed in vitamin D-induced and adenine/high-phosphate CKD models. These results suggest that the LIF–TYK2 signaling axis plays a pivotal role in CKD-associated vascular calcification and may serve as a future therapeutic target [133,134,135].

Type 1 diabetes (T1D) is an autoimmune disease characterized by the destruction of insulin-producing pancreatic β-cells, necessitating lifelong insulin replacement therapy. Its pathogenesis involves autoimmune-mediated β-cell destruction and is influenced by both genetic and environmental factors [136]. Inflammatory cytokines, particularly IFNs, play a key role in T1D development by inducing inflammatory responses, upregulating HLA class I molecules, triggering endoplasmic reticulum stress, and promoting β-cell apoptosis via the TYK2–STAT signaling axis and synergistic action with IL-1β [137,138,139,140,141,142,143,144,145]. Genetic studies have identified TYK2 as a risk factor of T1D [138,139]. TYK2 deficiency suppresses IL-12 signaling in CD8-positive T cells and inhibits cross-priming of cytotoxic CD8-positive T-bet T cells by resident dendritic cells in pancreatic lymph nodes. It also attenuates age-related β-cell inflammation. Administration of the selective TYK2 inhibitor BMS-986165 in autoimmune T1D mouse models significantly reduced the expansion of cytotoxic T-BET T cells, β-cell inflammation, and diabetes onset [140]. Notably, deucravacitinib was shown to preserve β-cell viability and function, protecting them from cytokine-induced damage, highlighting its potential for the prevention or treatment of T1D [141].

RNA interference screening revealed that T-cell acute lymphoblastic leukemia (T-ALL) cells depend on the TYK2–STAT1 signaling axis for survival. This dependency was observed across both primary T-ALL samples and established cell lines, and activation of the pathway was driven by gain-of-function mutations in TYK2 or IL-10 receptor signaling. TYK2-mediated STAT1 activation upregulates anti-apoptotic BCL2 expression, implicating the TYK2–STAT1–BCL2 survival axis in T-ALL pathogenesis [146,147,148,149]. Whole-exome sequencing of pediatric patients with multiple de novo leukemias identified germline TYK2 mutations located near the pseudokinase domain, potentially enhancing autophosphorylation and downstream STAT activation by disrupting the autoinhibitory conformation of the protein [147]. These findings suggest that germline-activating mutations in TYK2 may contribute to leukemogenesis and support the therapeutic targeting of this pathway. A novel TYK2 inhibitor, NDI-031301, demonstrated potent and selective cytotoxicity in human T-ALL cell lines by inducing apoptosis via the activation of mitogen-activated protein kinase (MAPK) pathways (extracellular signal-regulated kinase, ERK; Jun N-terminal kinase, JNK; and p38), with p38 MAPK playing a key role in cell death induction. In T-ALL xenograft mouse models, NDI-031301 significantly reduced the tumor burden and prolonged survival, with favorable tolerability. These results strongly support TYK2 inhibition as a promising therapeutic approach for T-ALL [150].

Multiple sclerosis (MS) is a prototypical demyelinating neurodegenerative disease of the central nervous system [151]. Autoimmune thyroid diseases share overlapping susceptibility profiles and immunopathogenic mechanisms with those of MS [152,153]. Integrated analyses of genome-wide association study (GWAS) and transcriptomic data have identified shared susceptibility genes between these two disorders, including JAK1, STAT3, IL2RA, HLA-DRB1, TLR3, and TYK2. These findings also point to the involvement of shared pathways such as JAK–STAT signaling, Th1 and Th17 cell differentiation, programmed death-ligand 1 (PD-L1) expression, and PD-1 checkpoint regulation in cancer immunity. Consequently, targeting TYK2 with agents such as deucravacitinib may be a novel therapeutic strategy for both MS and autoimmune thyroid disease [154].

**Table 3 ijms-26-09447-t003:** Diseases with increased TYK2 expression/activity and experimental support for TYK2 inhibition.

Disease	Evidence	Article
Alzheimer’s disease	CRISPR analysis identified TYK2 as a key mediator of cdsRNA-induced neurotoxicity, and in mouse models, TYK2 knockdown reduced total and pathogenic tau levels and improved gliosis.	König LE et al. [126] Kim J et al. [127]
Amyotrophic lateral sclerosis	CRISPR analysis identified TYK2 as a key mediator of cdsRNA-induced neurotoxicity, and in mouse models, TYK2 knockdown reduced total and pathogenic tau levels and improved gliosis.	Fröhlich A et al. [127] Kim J et al. [128]
Chronic kidney disease	In mouse models, TYK2 inhibition or genetic deletion reduced medial arterial calcification.	Alesutan I et al. [133]
Type 1 diabetes	Genetic studies have identified TYK2 as a risk factor for T1D, and in autoimmune T1D mouse models, the selective TYK2 inhibitor BMS-986165 suppressed β-cell inflammation and diabetes onset.	Onengut-Gumuscu S et al. [138] Mine K et al. [140]
T-cell acute lymphoblastic leukemia	RNA interference screening showed that T-ALL cells depend on the TYK2–STAT1–BCL2 axis for survival, and in mouse models, the novel TYK2 inhibitor NDI-031301 reduced tumor burden and prolonged survival.	Sanda T et al. [146] Akahane K et al. [150]
Multiple sclerosis	GWAS and transcriptomic analyses identified TYK2 as a susceptibility gene.	Thompson AJ et al. [151]
Autoimmune thyroid diseases	GWAS and transcriptomic analyses identified TYK2 as a susceptibility gene.	Munteis E et al. [152]

Abbreviations; CRISPR: clustered regularly interspaced short palindromic repeats, TYK2: tyrosine kinase 2, cdsRNA: cytosolic double-stranded RNA, T1D: type 1 diabetes, T-ALL: T-cell acute lymphoblastic leukemia, STAT: signal transducer and activator of transcription, BCL2: B-cell lymphoma 2, GWAS: genome-wide association study (studies).

## 3. Discussion

Currently, deucravacitinib is the only oral Jak inhibitor approved for plaque psoriasis. However, similar to JAK inhibitors, they have the potential to suppress a broad spectrum of immune responses mediated by IL-12, IL-23, and type I IFNs, thereby offering therapeutic benefits against various inflammatory diseases driven by these cytokines [7,8,9,10,11,12,13,14,15,16,17]. In this review, we examined various diseases for which case reports and clinical observations have suggested the efficacy of deucravacitinib, including DLE, SLE, AA, LP, PPP, PsA, SSc, IP, UC, CD, and CRMO [2,13,15,19,33,34,35,36,37,38,39,40,41,42,43,44,45,46,47,48,49,50,51,52,53,54,55,56,57,58,59,60,61,62,63,64,65,66,67,68,69,70,71,72,73,74,75,76,77,78,79,80,81,82,83,84,85,86,87,88,89,90,91,92,93,94,95,96,97,98,99,100,101,102,103,104,105,106,107,108,109,110,111,112,113,114,115,116,117,118,119,120,121]. A summary of the involved cytokines, JAK/STAT signaling pathways, and associated diseases is shown in Graphical Abstract. These diseases are commonly driven by cytokines, such as IL-23 and type I IFNs, in which TYK2 plays a central role. Thus, TYK2 inhibition may exert therapeutic effects by suppressing the activity of these inflammatory cytokines [2,13,15,19,33,34,35,36,37,38,39,40,41,42,43,44,45,46,47,48,49,50,51,52,53,54,55,56,57,58,59,60,61,62,63,64,65,66,67,68,69,70,71,72,73,74,75,76,77,78,79,80,81,82,83,84,85,86,87,88,89,90,91,92,93,94,95,96,97,98,99,100,101,102,103,104,105,106,107,108,109,110,111,112,113,114,115,116,117,118,119,120,121]. Among these, DLE, SLE, and AA are particularly associated with type I IFN signaling, whereas PsA, PPP, UC, CD, and CRMO are mainly driven by IL-23. In contrast, IP, SSc, and LP involve complex multifactorial inflammatory mechanisms, to which both type I IFN and IL-23 contribute [2,13,15,19,33,34,35,36,37,38,39,40,41,42,43,44,45,46,47,48,49,50,51,52,53,54,55,56,57,58,59,60,61,62,63,64,65,66,67,68,69,70,71,72,73,74,75,76,77,78,79,80,81,82,83,84,85,86,87,88,89,90,91,92,93,94,95,96,97,98,99,100,101,102,103,104,105,106,107,108,109,110,111,112,113,114,115,116,117,118,119,120,121]. Although deucravacitinib has not yet been approved for treating these conditions, ongoing clinical trials may expand its indications. Actually, in a phase 2 clinical trial involving patients with SLE, deucravacitinib demonstrated significant improvements not only in the primary endpoint, the SRI-4 response rate, but also across multiple secondary endpoints, including joint assessments and SLE-related biomarkers such as anti-dsDNA antibody titers and complement components C3 and C4, compared with placebo [46]. These findings suggest the potential of deucravacitinib as a promising therapy for SLE. However, although the differences versus placebo were statistically significant, the effect size was only moderate, and the trial did not specifically evaluate organ-specific endpoints [46]. Therefore, it remains unclear to what extent deucravacitinib can improve individual organ involvement and long-term outcomes in patients with SLE who present with internal organ damage, highlighting the need for further evidence. In addition to these diseases in which deucravacitinib may be effective, experimental studies using mouse models or human cells have demonstrated increased TYK2 expression and the therapeutic potential of TYK2 inhibition in Alzheimer’s disease, T1D, CKD, T-ALL, and MS [122,123,124,125,126,127,128,129,130,131,132,133,134,135,136,137,138,139,140,141,142,143,144,145,146,147,148,149,150,151,152,153,154]. These findings support the potential clinical applications of deucravacitinib in a broad range of diseases.

In the JAK/STAT signaling pathway, TYK2 is known to associate not only with the signaling of IL-12, IL-23, and type I IFN but also with multiple cytokine receptor families, including the IL-6, IL-10, and IL-13 families. Upon binding of cytokines such as IL-6, IL-10, or IL-13 to their respective receptors, receptor subunit dimerization occurs, leading to phosphorylation and activation of TYK2 in conjunction with its partner kinases, JAK1 or JAK2 [1,2,3,4,5,6,15,16,17,155]. However, studies in TYK2-deficient mice have demonstrated that while TYK2 is essential for responses to IL-12, IL-23, and type I IFN, it is not strictly required for IL-6 or IL-10 signaling [149,156,157]. Similar findings have been observed in primary human cell cultures lacking TYK2, where signal transduction mediated by IL-6, IL-10, and IL-13 is largely preserved [158]. These findings suggest that although TYK2 interacts with multiple cytokine receptors and contributes to downstream activation, its role in IL-6, IL-10, and IL-13 signaling may be limited [149,156,157,158]. TYK2 possesses both catalytic activity and kinase-independent scaffold functions. In fact, it has been proposed that TYK2 may be essential for the surface expression of cytokine receptors such as IFN-α and IL-10 receptor through mechanisms independent of its enzymatic activity [159]. This indicates that despite being part of the JAK family, TYK2 may play a more complex and nuanced role in cytokine signaling than previously appreciated. Nevertheless, these insights are primarily derived from mouse models and in vitro human cell studies [149,155,156,157,158,159]. It remains unclear how TYK2 inhibition with agents such as deucravacitinib affects cytokine signaling in diseases characterized by excessive IL-6 activity, such as SSc, or IP, or IL-13–driven conditions like atopic dermatitis [2,15,77,78,79,80,81,82,83,84,85,86,104,105,106,107,108,109,110,111,112,160]. Further accumulation of clinical experience, along with mechanistic and translational research, is required to clarify these effects.

Current psoriasis treatment guidelines recommend that biologics and deucravacitinib should be discontinued in patients with a history of malignancy and advise against its use for at least five years following cancer remission [161]. However, emerging evidence suggests a more nuanced role of TYK2 in oncogenesis. Aberrant TYK2 activation has been implicated in T-ALL and in vitro studies have demonstrated the potential antitumor effects of TYK2 inhibitors in T-ALL models [146,147,148,149,150]. Moreover, TYK2 overexpression has been observed not only in hematologic malignancies such as non-Hodgkin lymphoma, but also in solid tumors such as lung cancer, suggesting a potential role for TYK2 inhibitors in cancer therapy [162,163]. Given these findings, while the current guidelines urge caution in prescribing TYK2 inhibitors to patients with a history of malignancy, clinical decision-making in practice may benefit from a more individualized approach [161,162,163]. Factors such as the type and treatment course of the malignancy, severity of psoriasis, and patient preferences should be carefully considered. Under close monitoring, TYK2 inhibition can potentially be considered a therapeutic option, even in such populations [146,147,148,149,150,161,162,163]. As additional clinical data and case studies accumulate, the evidence base is expected to expand, supporting the safe and effective use of TYK2 inhibitors in patients with histories of malignancy.

This review examined the therapeutic potential of deucravacitinib, which is currently approved for the treatment of psoriasis, in a wide range of inflammatory diseases. The conditions discussed include those under active clinical trials and those for which anecdotal evidence of improvement has been reported, particularly in patients with comorbid psoriasis. The advantages of deucravacitinib include its broad immunosuppressive effects and favorable safety profile, making it an appealing treatment option without major concerns regarding adverse events. If its indications are expanded, it could offer significant clinical benefits to a broader patient population [164].

Further research is warranted to explore the interactions between TYK2 and cytokines, such as IL-6, IL-10, and IL-13, which are currently thought to be less dependent on TYK2 signaling. In addition, elucidating the mechanisms by which TYK2 inhibition exerts its efficacy in promising inflammatory diseases is critical for optimizing its clinical application.

## 4. Conclusions

Deucravacitinib, a selective oral TYK2 inhibitor, has demonstrated robust efficacy and a favorable safety profile in psoriasis and holds considerable promise for the treatment of a broad spectrum of immune-mediated diseases. Accumulating evidence from clinical trials, real-world studies, and mechanistic research indicates potential therapeutic benefits in conditions driven by the type I IFN and IL-23 pathways, such as discoid lupus erythematosus, systemic lupus erythematosus, alopecia areata, psoriatic arthritis, and chronic recurrent multifocal osteomyelitis. In addition, its efficacy has only been suggested by case series, and while the evidence remains preliminary and requires cautious interpretation, deucravacitinib may still hold promise for complex, multifactorial diseases such as systemic sclerosis, interstitial pneumonia, and lichen planus. Furthermore, the preliminary findings suggest possible applications in neurodegenerative disorders, metabolic and fibrotic diseases, and certain malignancies.

Although its current indications remain limited to psoriasis, the breadth of cytokine signaling modulated by TYK2 inhibition supports its potential for therapeutic expansion. Future research should focus on clarifying the role of TYK2 in cytokine pathways that are less dependent on its activity, optimizing dosing strategies for disease-specific contexts, and conducting large-scale clinical trials to validate the efficacy and safety of under diverse conditions. With continued investigation, deucravacitinib may emerge as a versatile treatment option capable of addressing unmet needs in a wide range of inflammatory and immune-mediated disorders.

## Data Availability

Data concerning this article may be requested from the corresponding author for reasonable reasons.

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
