# Peer review of "Review of Promising Off-Label Use of Deucravacitinib"

_ijms, 2025, doi:10.3390/ijms26199447_

Round 1
Reviewer 1 Report
Comments and Suggestions for Authors
This manuscript offers a broad and timely narrative review of the potential off‑label uses of the selective TYK2 inhibitor deucravactinib across many diseases.
You already hint that lower BMI associate with better responses in psoriasis; synthesize this explicitly and cite responder‑prediction work to guide selection. In particular, consider adding Cureus (doi: 10.7759/cureus.86044) and the stratified J Dermatol (doi: 10.1111/1346‑8138.17601) as practical resources on patient selection and response predictors.
Taken together, this is very interesting paper—timely, clinically useful, and I believe after these fixes it will read much stronger.
Author Response
#Reviewer 1
Thank you very much for your detailed and thoughtful review. Below, we provide our point-by-point responses to the reviewers’ comments. We would be grateful for your confirmation.
Comment 1-1: This manuscript offers a broad and timely narrative review of the potential off‑label uses of the selective TYK2 inhibitor deucravactinib across many diseases.
You already hint that lower BMI associate with better responses in psoriasis; synthesize this explicitly and cite responder‑prediction work to guide selection. In particular, consider adding Cureus (doi: 10.7759/cureus.86044) and the stratified J Dermatol (doi: 10.1111/1346‑8138.17601) as practical resources on patient selection and response predictors. Taken together, this is very interesting paper—timely, clinically useful, and I believe after these fixes it will read much stronger.
Our Reply 1-1: Thank you very much for your thorough and thoughtful review. In accordance with your suggestions, we have added Cureus (doi: 10.7759/cureus.86044) and J Dermatol (doi: 10.1111/1346-8138.17601) as new references 25 and 26. We have also incorporated additional content noting that while patients with lower BMI or older age tend to show better treatment responsiveness, younger patients are more likely to achieve PASI clear. This has further strengthened our description of the therapeutic efficacy of deucravacitinib in psoriasis. These revisions are shown in Line 104-117 in red characters. We would greatly appreciate your kind confirmation.
In addition, we have updated the reference numbering to accommodate newly added citations and made minor revisions to the Conclusion to reflect the content changes. All modifications are highlighted in red in the revised manuscript. We appreciate your kind consideration.
Reviewer 2 Report
Comments and Suggestions for Authors
I appreciated having the opportunity to review the paper about the expanding therapeutic potential of deucravacitinib, a selective oral TYK2 inhibitor currently approved for psoriasis.
I just have some suggestions that I think can help improve the paper overall.
The review sometimes seems like an accumulation of case reports and trial outcomes without enough subjective comments.It could be very useful and clarifying to provide summary tables or schematic figures that categorize diseases.
While the Phase II data on te use of the drug in LES are encouraging, the effect sizes were modest and the trial was not made for organ-specific endpoints. Please underline this aspect in the discussion.
In systemic sclerosis and interstitial pneumonia, only case-level data are presented. The text should stress that evidence remains preliminary as this aspect seems misleading in the paper right now.
The IBD section could include the most recent updates on risankizumab and guselkumab approvals as well to make the paper more recent in its content.
Author Response
#Reviewer 2
Thank you very much for your detailed and thoughtful review. Below, we provide our point-by-point responses to the reviewers’ comments. We would be grateful for your confirmation.
Comment 2-1: I appreciated having the opportunity to review the paper about the expanding therapeutic potential of deucravacitinib, a selective oral TYK2 inhibitor currently approved for psoriasis. I just have some suggestions that I think can help improve the paper overall. The review sometimes seems like an accumulation of case reports and trial outcomes without enough subjective comments. It could be very useful and clarifying to provide summary tables or schematic figures that categorize diseases.
Our reply 2-1: Thank you for the valuable suggestion. We have created three new tables. Table 1 summarizes diseases in which case reports suggest a therapeutic benefit of deucravacitinib. Table 2 compiles diseases for which therapeutic potential has been suggested and clinical trials of deucravacitinib are currently ongoing. Table 3 summarizes diseases in which preclinical/experimental studies indicate therapeutic potential. We would appreciate your review.
Comment 2-2: While the Phase II data on te use of the drug in LES are encouraging, the effect sizes were modest and the trial was not made for organ-specific endpoints. Please underline this aspect in the discussion.
Our reply 2-2: Thank you for the insightful comment. We have expanded the Discussion to note that, although the phase II data in SLE are encouraging, the observed effect size is moderate and the trial was not designed around organ-specific endpoints, focusing instead on articular symptoms and laboratory parameters (Lines 614–624, in red).
Comment 2-3: In systemic sclerosis and interstitial pneumonia, only case-level data are presented. The text should stress that evidence remains preliminary as this aspect seems misleading in the paper right now.
Our reply 2-3: Thank you for the insightful comment. Regarding systemic sclerosis and interstitial lung disease, we emphasize that the current evidence consists only of case-level observations suggesting potential benefit; therefore, the evidence base remains preliminary (additions at Lines 349 and 458).
Comment 2-4: The IBD section could include the most recent updates on risankizumab and guselkumab approvals as well to make the paper more recent in its content.
Our reply 2-4: Thank you for the constructive suggestion. We have now cited the most recent clinical trial publications on risankizumab and guselkumab for inflammatory bowel disease (IBD) and updated the text to summarize their latest clinical outcomes. These revisions are marked in red at Lines 368–398.
In addition, we have updated the reference numbering to accommodate newly added citations and made minor revisions to the Conclusion to reflect the content changes. All modifications are highlighted in red in the revised manuscript. We appreciate your kind consideration.
Reviewer 3 Report
Comments and Suggestions for Authors
Thank you for this thorough review. It is helpful to gather information on other possible future indications or off-label uses of deucravacitinib. Please find my comments attached:
-
Line 85. In the POETYK PSO-1 clinical trial, PASI 90 at week 24 was 42.2%, which is well below 50%. I encourage the authors to be more precise when describing trial results.
-
Discussion. The first sentence in line 525 should be corrected. The authors likely intend to state that deucravacitinib is the only JAK inhibitor approved for the treatment of psoriasis.
-
Suggestion. Consider creating a table summarizing all case reports, including the different diseases, number of patients, patient characteristics, responses, and ongoing clinical trials.
Author Response
#Reviewer 3
Thank you very much for your detailed and thoughtful review. Below, we provide our point-by-point responses to the reviewers’ comments. We would be grateful for your confirmation.
Comment 3-1: Thank you for this thorough review. It is helpful to gather information on other possible future indications or off-label uses of deucravacitinib. Please find my comments attached:Line 85. In the POETYK PSO-1 clinical trial, PASI 90 at week 24 was 42.2%, which is well below 50%. I encourage the authors to be more precise when describing trial results.
Our reply 3-1: Thank you for the suggestion. We have revised the manuscript to state that approximately 42% of patients achieve PASI 90. This revision is shown in line 86 in red characters.
Comment 3-2: Discussion. The first sentence in line 525 should be corrected. The authors likely intend to state that deucravacitinib is the only JAK inhibitor approved for the treatment of psoriasis.
Our reply 3-2: Thank you for the comment. We have revised the manuscript to state that deucravacitinib is currently the only approved oral JAK inhibitor (TYK2) for plaque psoriasis. This revision is shown in line 599 in red characters.
Comment 3-3: Suggestion. Consider creating a table summarizing all case reports, including the different diseases, number of patients, patient characteristics, responses, and ongoing clinical trials.
Our reply 3-3: Thank you for the valuable suggestion. We have created three new tables. Table 1 summarizes diseases in which case reports suggest a therapeutic benefit of deucravacitinib. Table 2 compiles diseases for which therapeutic potential has been suggested and clinical trials of deucravacitinib are currently ongoing. Table 3 summarizes diseases in which preclinical/experimental studies indicate therapeutic potential. We would appreciate your review.
In addition, we have updated the reference numbering to accommodate newly added citations and made minor revisions to the Conclusion to reflect the content changes. All modifications are highlighted in red in the revised manuscript. We appreciate your kind consideration.